# Detection of Circulating and Disseminated Tumor Cells and Their Prognostic Value under the Influence of Neoadjuvant Therapy in Esophageal Cancer Patients

**DOI:** 10.3390/cancers14051279

**Published:** 2022-03-01

**Authors:** Florian Richter, Christian Röder, Thorben Möller, Jan-Hendrik Egberts, Thomas Becker, Susanne Sebens

**Affiliations:** 1Department of General, Visceral-, Thoracic-, Transplantation- and Pediatric Surgery, University Medical Center Schleswig-Holstein (UKSH), Campus Kiel, 24105 Kiel, Germany; florian.richter@uksh.de (F.R.); thorben.moeller@uksh.de (T.M.); thomas.becker@uksh.de (T.B.); 2Institute for Experimental Cancer Research, Kiel University (CAU) and University Medical Center Schleswig-Holstein (UKSH), Campus Kiel, 24105 Kiel, Germany; c.roeder@email.uni-kiel.de; 3Department of Surgery, Israelitisches Krankenhaus, 22297 Hamburg, Germany; j.egberts@ik-h.de

**Keywords:** liquid biopsy, esophageal carcinoma, CTC, DTC, CK20, DEFA5

## Abstract

**Simple Summary:**

Esophageal cancer (EC) has a poor prognosis and a high mortality rate. This study investigated the expression of CK20 and DEFA5, markers being associated with circulating (CTC) and disseminated tumor cells (DTC), in blood and bone marrow (BM) of EC patients, and correlated positivity rates with clinical data to assess the prognostic impact. Both markers were detected in blood and BM of EC patients and the control cohort so that a cut-off value was determined to define marker positivity for correlation with clinical parameters. CK20 and DEFA5 positivity in liquid biopsies of EC patients did not correlate with overall survival (OS). However, CK20 positivity in BM and DEFA5 negativity in blood were associated with reduced OS in patients without neoadjuvant therapy. In patients with neoadjuvant therapy, DEFA5 positivity in BM was associated with improved OS, pointing to the potential of DEFA5 as a prognostic biomarker in liquid biopsies of EC patients.

**Abstract:**

Detection of circulating (CTC) or disseminated tumor cells (DTC) are correlated with negative prognosis in esophageal cancer (EC) patients. In this study, DTC- and CTC-associated markers CK20 and DEFA5 were determined by RT-PCR in EC patients and correlated with clinical parameters to determine their prognostic impact. The blood and bone marrow (BM) of 216 EC patients after tumor resection with or without neoadjuvant therapy and as control blood samples from 38 healthy donors and BM from 24 patients with non-malignant diseases were analyzed. Both markers were detected in blood and BM of EC patients and the control cohort. A cut-off value was determined to define marker positivity for correlation with clinical data. CK20 expression was detected in 47/206 blood samples and in 49/147 BM samples of EC patients. DEFA5 positivity was determined in 96/206 blood samples and 98/147 BM samples, not correlating with overall survival (OS). However, CK20 positivity in BM and DEFA5 negativity in blood were associated with reduced OS in EC patients without neoadjuvant therapy, while in patients with neoadjuvant therapy DEFA5 positivity in BM was associated with improved OS. Overall, our study suggests DEFA5 as a prognostic biomarker in liquid biopsies of EC patients which requires further validation.

## 1. Introduction

According to global cancer statistics, esophageal cancer (EC) represents the sixth highest cause of cancer-related deaths with an ever-increasing incidence [1,2]. The five-year overall survival rate is only 20% for this type of cancer, which is mainly due to the advanced tumor stage at the time of diagnosis [3,4]. Due to a mortality rate of up to 90%, it belongs to the predominant tumor entities with a poor prognosis [5]. Thus, despite the ongoing development of diagnostic measures such as endoscopy, computed tomography (CT), and positron emission tomography (PET), more than 60% of EC patients are diagnosed at an advanced tumor stage. This can be explained by the lack of symptoms in the early stage of the disease [6]. Clinical data show that in around 50% of patients, who were operated and showed no evidence of local or distant metastasis, a progression of disease occurred within 12 months after surgery [7]. This is thought to be due to a clinically undetectable so-called minimal residual disease (MRD), which may be circulating tumor cells (CTC) or disseminated tumor cells (DTC) [8]. Both entities of tumor cells can originate from the same primary tumor and can be detected either in the blood as CTC or in the bone marrow and other secondary sites such as the liver or lung as DTC [9,10]. 

Preoperative diagnostics of esophageal carcinomas for the assessment of tumor extension and clarification of possible metastases are made on the basis of the results from different imaging modalities such as endoscopy, CT, and PET [11]. Postoperative staging is assigned to the clinical stages I–IV in accordance with the TNM classification of the UICC. This classification allows for a good prediction regarding metastasis-free and overall survival but the problem remains the detection of early recurrence due to low sensitivity of minor lesions.

Analyses of “liquid biopsies”, which represent a simple and non-invasive sampling technique of cells and cell-derived biomarkers (exosomes, DNA, RNA) from peripheral blood or bone marrow, is a powerful tool providing a more sensitive and precise approach allowing detection of CTC and DTC, respectively. This would result in a more precise prognostic staging and, moreover, when applied in a long-term follow-up, could allow a more accurate therapy monitoring in EC patients and early detection of relapses [12,13]. Both CTC and DTC have the potential to be used as predictive and prognostic biomarkers to monitor therapeutic effects in cancer patients [14,15,16]. Previous studies which analyzed CTC in peripheral blood of EC patients identified them as an independent prognostic marker [17,18,19]. In addition, a correlation between the presence of DTC in bone marrow and the increased risk of metastasis after tumor resection was demonstrated in EC patients [20].

Until now, a multitude of techniques has been developed to detect CTC and DTC, being principally based on their enrichment and detection on either a cellular or molecular level [8,21,22]. In this study, the established epithelial cell marker cytokeratin-20 (CK20)- as well as α-defensin 5 (DEFA5)-mRNA expression levels were determined as possible CTC and DTC surrogate markers in EC patient’s blood and bone marrow mononuclear cell (MNC) fractions. MNC enrichment by Ficoll density centrifugation represents a long-established technique for depleting possibly CK20-positive granulocytes from whole blood samples [23], also enriching CTC and DTC as demonstrated in numerous studies with clinical samples as well as in preanalytical validation experiments with spiked tumor cells in whole blood [24]. In the present study, gene expression was analyzed by using reverse-transcription real-time-polymerase chain reaction (RT-PCR) assays. 

CK20 is a low-molecular-weight member of the cytokeratin family of cytoskeletal proteins expressed in epithelial cells. This intermediate filament protein is involved in cell structure and differentiation. Although being a general epithelial cell marker, several studies identified an association between CK20-positive cells in blood and the presence of gastrointestinal tumors [25,26,27].

Defensins are anti-microbial peptides of the innate immune system and can be detected in the secretory granules of neutrophils and are also expressed in the surface epithelium of various organs, especially in the small and large intestine, in the lung as well as in the esophagus. DEFA5 is known for its high expression in secretory Paneth cells, which are found abundantly in the esophageal epithelium, but also in colon cancer, ovarian, endometrium, and pulmonary carcinomas [28,29,30]. One study demonstrated a correlation between elevated plasma levels of α-defensins 1–3 and lymphatic and hepatic metastasis in colorectal cancer (CRC) [31]. Moreover, we recently demonstrated elevated DEFA5 mRNA expression in blood-derived MNC of CRC patients compared to patients with benign diseases of this organ system and healthy donors [32]. However, despite some reports on the function of DEFA5 in the regulation of cell growth and apoptosis, its role in tumorigenesis is not yet well understood [33,34,35]. The fact that DEFA5 can negatively affect E-cadherin expression in esophageal squamous cells suggests that it may promote the dissemination of EC cells from the primary context and may thereby play a crucial role in EC progression [36]. Based on our previous findings on CK20 mRNA expression in liquid biopsies in a small cohort of EC patients and DEFA5 mRNA expression in blood samples of CRC patients, the present study aimed to validate our findings and to investigate whether CK20 and DEFA5 can be used as prognostic markers in liquid biopsies of EC patients.

CK20 and DEFA5 mRNA expression was determined in MNC fractions from blood and bone marrow taken pre-operatively before tumor resection of a cohort of 216 EC patients. To assess the prognostic value of these markers, CK20 and DEFA5 positivity or negativity and semi-quantitative expression values were correlated with various clinical and follow-up parameters. Moreover, it was investigated whether pre-operative CTC/DTC detection might increase the sensitivity and accuracy in the diagnosis of EC and might thereby improve pre-operative staging or strategy for the treatment of EC patients in the presence of CTC/DTC.

## 2. Results

### 2.1. Characterization of the Analyzed EC Patient Cohort

The patient collective comprised 216 patients with a median age of 63 years (range 29–84 years). The 5-year OS rate was 38% and the median overall survival was 27.5 months (range: 1–244 months). Of the 216 patients, 182 (84.3%) were male and 34 (15.7%) were female. One hundred and sixty-two (75%) patients presented with adenocarcinoma (AC), 54 (25%) with squamous cell carcinoma (SCC). Furthermore, 115 (53.2%) of the patients received neoadjuvant chemotherapy, whereas 101 (46.8%) of the patients were untreated before surgery. Adjuvant treatment was administered in 57 (26.4%) patients in our cohort, whereas 138 (63.9%) patients did not receive any postoperative adjuvant treatment. For 21 (9.7%) patients, it was not known whether adjuvant therapy was given. As expected, we found a significant and strong correlation between UICC-stage and 5-year OS (*p* < 0.001). Furthermore, also the TNM-categories showed a highly significant correlation with the 5-year OS: a high pT category and positive lymph node (pN) status predicted a worse 5-year OS with high significance (*p* < 0.001). Additionally, a significant correlation between the 5-year OS and the occurrence of distant metastasis could be shown (*p* = 0.001). Expectedly, also the patients’ age at the time of surgery showed a significant correlation with their survival. All other variables analyzed did not show any correlation with the survival rate. All data are summarized in Table 1.

### 2.2. Quantitative Analysis of CK20 and DEFA5 mRNA Expression by RT-PCR

We demonstrated earlier in a smaller cohort of EC patients that the epithelial marker CK20 can be utilized for detection of CTC in the blood and DTC in the bone marrow of EC patients [19]. Other studies have described associations between the DEFA5 expression and gastrointestinal tumorigenesis [29]. In addition, DEFA5 was shown to promote a reduction in E-cadherin in esophageal epithelial cells which might be a mechanism contributing to EC dissemination from the primary tumor [36]. To investigate whether CK20 and DEFA5 expression can be used as prognostic and predictive biomarkers in blood and bone marrow of EC patients, CK20 and DEFA5 quantitative RT-PCR assays were performed with RNA prepared from 209 blood and 147 bone marrow samples of 216 EC patients. As a control, 38 blood samples from healthy donors and 24 bone marrow samples from patients with non-malignant diseases (see Section 3) were analyzed (Figure 1A,B).

In the 209 blood samples of EC patients examined, RT-PCR showed positive CK20 detection in 84 patients (positivity rate: 40.2%) and 125 blood samples have been tested negative. Analyzing blood samples for DEFA5 expression, 148 samples were positive (70.8%) and 61 samples were negative. Examination of the bone marrow samples of EC patients revealed a CK20 positivity in 60 patients (40.8%) and 87 bone marrow samples presented negative. In contrast, DEFA5 positivity was detected in all bone marrow samples examined. In the control cohort, 24 blood samples examined were positive for CK20, while 14 samples were negative for CK20. For DEFA5 we found a positive detection in 22 individuals and 16 samples were negative. Examination of the bone marrow samples of our control cohort, the detection of CK20 was found in 10 patients and of DEFA5 in 22 of the 24 samples (Figure 1A,B). CK20 detection was similar in both compartments with 40.8% in the bone marrow and 40.2% in the blood. The situation was different for DEFA5, where we detected a higher positive detection rate in the bone marrow than in the blood. Thus, both markers examined were also detected in the control cohort. However, it has to be taken into account that all RT-PCR analyses were performed in triplicate with the samples from the EC patients being more often triple or at least double positive for either marker than the control cohort.

### 2.3. Definition of a Diagnostic Cut-Off Threshold of CK20 and DEFA5

The RT-PCR analysis of CK20 and DEFA5 showed a pronounced variability of mRNA expression in either cohort ranging from lower to rather high c_T_ values, the latter indicating very low gene expression.

Thus, in the EC cohort, the calculated CK20 expression units (EU) in the blood ranged from 0.7 EU to 35.6 EU, and in the bone marrow from 3.3 EU to 56.3 EU. For DEFA5, expression levels in the blood ranged from 0.04 EU to 460 EU and in the bone marrow from 138.78 EU to 8315.81 EU.

In the control cohort, expression levels in blood ranged from 2.46 EU to 15.15 EU for CK20 and from 2.15 EU to 13.47 EU for DEFA5. Examination of bone marrow samples showed expression units for CK20 in the range of 5.26 EU to 35.69 EU and for DEFA5 of 81.42 EU to 2384.78 EU. Even though the cohort of EC patients comprised more blood and bone marrow samples with clearly higher expression levels of either marker compared to the control samples, a high variability of the expression values was determined, so it was decided to determine cut-off values for sample positivity, as described below, which are indicated as red horizontal lines in Figure 1A,B. 

In order to estimate the threshold expression levels of CK20 and DEFA5 corresponding to the highest sensitivity and specificity of these biomarkers, cut-off values were determined by calculation of the Youden indices after receiver operating characteristics (ROC) analysis. The Youden ’s index J was calculated by the equation: J = sensitivity + specificity − 1 yielding a cut-off value of 4.64 EU for CK20 in blood samples and a cut-off value of 6.06 EU in bone marrow samples of EC patients (Figure 1A). For DEFA5, cut-off values of 4.26 EU in blood samples and of 599.3 EU in bone marrow samples were calculated (Figure 1B).

### 2.4. Semi-Quantitative Analysis of CK20 and DEFA5 mRNA Expression by RT-PCR

Considering the RT-PCR results and including the calculated cut-off values, the following detection rates were determined in the 216 EC patients. As shown in Table 2, 47 of 209 (22.5%) blood samples tested positive for CK20 and in 49 of 147 (33.3%) bone marrow samples CK20 expression could also be detected. DEFA5 positivity could be determined in 96 of 209 blood samples (45.9%) and 98 of 147 bone marrow samples (66.7%). 

These data indicate that the majority of EC patients are negative for CK20 in blood and bone marrow, while DEFA5 positivity is more frequently found in blood and bone marrow.

### 2.5. Correlation of CK20 and DEFA5 in Liquid Biopsies of EC Patients with Clinical Parameter

The subsequent analysis aimed at showing a correlation between detection of CK20 or DEFA5 expression in the blood or bone marrow of EC patients with pre-determined clinical parameters. In our collective, gender, age, different tumor entities, and UICC stage did not correlate with CK20 or DEFA5 expression in blood or bone marrow samples of EC patients. Moreover, no correlation was observed for the detection of either marker and neoadjuvant treatment or adjuvant therapy (Table 3). 

### 2.6. Correlation of CK20 and DEFA5 Expression in Blood and Bone Marrow with EC Patients’ Survival

Using the calculated cut-off values to re-evaluate detection rates of CK20 and DEFA5, the correlation of the positivity or negativity of either biomarker with the overall survival was analyzed by Kaplan–Meier estimation. As shown in Figure 2, no significant correlation between CK20 (*p* = 0.291) or DEFA5 (*p* = 0.420) positivity in blood samples and OS could be determined (Figure 2A,B). Interestingly, the survival curves of patients with CK20 or DEFA5 positivity in the blood seemed even more favorable than those of the biomarker-negative cases; however, the log-rank tests did not reveal any statistical differences between the two groups. Likewise, no correlation between CK20 (*p* = 0.198) or DEFA5 (*p* = 0.420) positivity in bone marrow and OS was determined (Figure 2C,D). Overall, these data do not suggest CK20 and DEFA5 expression neither in blood nor in bone marrow as a prognostic biomarker in EC patients.

### 2.7. Correlation between CK20 or DEFA5 Detection in Blood or Bone Marrow and Survival in EC Patients without Neoadjuvant Chemotherapy

Since our EC patient cohort comprises patients without and with neoadjuvant therapy, it was next investigated whether the positivity of CK20 or DEFA5 can serve as a prognostic biomarker in EC patients without neoadjuvant therapy. From 216 EC patients studied here, 101 patients did not receive neoadjuvant therapy, from which 98 blood and 65 bone marrow samples could be analyzed.

It was shown that there was no significant correlation between CK20 mRNA detection in blood and OS (*p* = 0.70) and similarly, no correlation could be shown between OS and detection of DEFA5 in bone marrow samples (*p* = 0.19) in the group of patients who had not received neoadjuvant therapy.

However, detection of CK20 in bone marrow was highly significantly correlated with a worse OS (*p* = 0.006). While CK20-negative patients (*n* = 41) had a median overall survival of 80 months, CK20-positive patients (*n* = 24) had a median survival of 20 months. The Kaplan–Meier plot demonstrates that the 5-year OS rate is significantly lower in patients without neoadjuvant chemotherapy and CK20-positive bone marrow than in patients with CK20-negative bone marrow (18% versus 54%) (Figure 3A). Moreover, and interestingly, there was a significant positive correlation between OS and the detection of DEFA5 in blood (*p* = 0.047). After five years, 48% of the DEFA5-positive patients (*n* = 43), but only 30% of the DEFA5-negative patients (*n* = 55), were still alive (Figure 3B). Thus, these data indicate that CK20 positivity in bone marrow is a negative prognostic marker, while DEFA5 positivity in blood is a positive prognostic biomarker in EC patients who did not receive neoadjuvant therapy. 

### 2.8. Correlation of CK20 or DEFA5 Positivity in Blood or Bone Marrow and Survival in EC Patients with Neoadjuvant Chemotherapy

To investigate whether the expression of CK20 or DEFA5 may serve as a prognostic biomarker for survival in the subcohort of EC patients with pre-operative therapy, detection rates in 111 blood samples and 82 bone marrow samples of 115 patients who underwent neoadjuvant treatment were correlated with OS. No significant correlation between CK20 detection in blood (*p* = 0.30) or bone marrow (*p* = 0.46) and OS could be observed in this patient cohort. Similarly, the detection of DEFA5 expression in blood was not associated with OS in patients with neoadjuvant therapy (*p* = 0.47).

However, analysis of our EC patients with neoadjuvant therapy showed a highly significant correlation between detection of DEFA5 in bone marrow and a longer OS (*p* = 0.001). The 5-year overall survival for DEFA5-positive patients was 47% compared to 16% for patients with a DEFA5-negative bone marrow. Of the DEFA5-negative patients in this collective, half of the patients had already died after 16 months in comparison to 43 months for DEFA5-positive patients (Figure 4). Overall, these data suggest DEFA5 mRNA in bone marrow as a positive prognostic biomarker in EC patients with neoadjuvant therapy.

### 2.9. Multivariate Analyses

In order to investigate whether detection of CK20 or DEFA5 expression in liquid biopsies are independent prognostic factors for EC patients with or without neoadjuvant therapy, the stepwise inclusion and the likelihood quotient (LQ) method were used as part of the Cox regression model for multivariate analyses. The calculation merely included variables that had a significant *p*-value following the log-rank test as part of the univariate analysis.

Thus, in the group of EC patients without neoadjuvant treatment, CK20 positivity in bone marrow and DEFA5 positivity in blood samples were included in the multivariate analysis. In the cohort of patients with neoadjuvant therapy, detection of DEFA5 in bone marrow was proven to be significant in the univariate analysis and therefore considered in the multivariate analysis. Since the univariate analysis showed a significance for the UICC stage in both groups, this variable was also included in this multivariate analysis. 

As seen in Table 4, CK20 positivity in the bone marrow of EC patients without neoadjuvant therapy was identified as an independent predictor of worse OS (HR 2.28; 95% CI 1.24–4.22; *p* = 0.008). 

In patients with neoadjuvant treatment multivariate analysis revealed DEFA5 as an independent prognostic marker for OS (HR 0.46; 95% CI 0.26–0.80; *p* = 0.006). Here, DEFA5 detection in bone marrow was shown to have a protective effect on overall survival. Finally, a higher UICC stage was determined as an independent marker of worse OS in both groups (HR 1.60; 95% CI 1.19–2.17; *p* = 0.002/HR 1.53; 95% CI 1.10–2.13; *p* = 0.012) (Table 4).

## 3. Materials and Methods

### 3.1. Patient Cohort

In total, the patient collective consisted of 216 EC patients. All patients underwent surgical tumor resection in the Department of General Surgery and Thoracic Surgery, University Hospital Schleswig Holstein (UKSH), Campus Kiel between 2000 and 2018 and were histologically verified as esophageal carcinoma at the Institute of Pathology UKSH, Campus Kiel. The median follow-up period was 48 months (1–244 months). Patients with squamous cell (*n* = 54) and adenocarcinoma (*n* = 162) were included. 

The study was approved by the local ethics committee of the Medical Faculty, Kiel University (reference no. A110/99). All patients gave written informed consent before inclusion in the study. Classification of the pathological tumor stage was conducted at the Institute of Pathology, UKSH Campus Kiel according to the edition of the TNM classification. Clinical data were obtained from the clinical patient files and the clinical research database of the oncological biobank BMB-CCC of the Medical Faculty of the University of Kiel. Clinical and follow-up data were then analyzed in correlation to the CK20 and DEFA5 expression data obtained by RT-PCR assays as described below. Only patients with complete clinical data were considered for further analysis.

### 3.2. Control Group

The control collective (*n* = 62 individuals) consisted of 38 healthy volunteers from whom peripheral venous blood samples (*n* = 38) were obtained. The volunteers were randomly recruited and not age/sex matched. Part of this collective was already utilized and described in a previous study [37]. Furthermore, bone marrow samples were collected from 24 patients suffering from non-malignant diseases including one patient with liver cysts, three patients from whom parts of multiorgan donations were obtained, five patients suffering from pulmonary hematoma, three patients with hepatic hemangioma, three patients suffering from focal nodular hyperplasia of the liver, five patients with sigmoid diverticulitis and four patients with chronic pancreatitis. Informed written consent for participation in the study was obtained from all individuals of the control cohort and investigation of the samples was covered by the same approval of the local ethics committee as for cancer patients.

### 3.3. Sample Collection, Isolation of RNA and RT-PCR

Directly before surgery, 10 mL of bone marrow blood was aspirated from the spina iliaca anterior under general anesthesia subsequent to a small cutaneous incision. Venous blood (20 mL) was taken in parallel from a central venous line of patients undergoing surgery. Blood samples of healthy volunteers were drawn from the medial cubital vein. Lithium heparin was used as an anticoagulant. Since it was shown earlier that granulocytes are capable of expressing both CK20 [23] and DEFA5 [29] and to increase detection sensitivity and specificity, blood and bone marrow samples were depleted of granulocytes by density-centrifugation through a Ficoll-Hypaque cushion (GE Healthcare, Freiburg, Germany) according to the manufacturer’s recommendation. This led to fractions of MNC also containing epithelial CTC or DTCs, which were further washed in phosphate-buffered saline (PBS) and counted. The MNC fractions were subsequently lysed with RNAPureTM reagent (VWR Peqlab, Darmstadt, Germany) and total RNA preparation was carried out according to the manufacturer’s protocol. RNA concentrations were measured by a NanoDrop 2000c Spectrophotometer (VWR Peqlab, Darmstadt, Germany). RNA integrity was verified using a Bioanalyzer 2100 instrument (Agilent Technologies, Böblingen, Germany).

cDNA was obtained by reverse transcription of 3 μg total RNA (Maxima First Strand cDNA Synthesis Kit, Thermo Fisher Scientific, Darmstadt, Germany) according to the manufacturer’s protocol. Realtime qPCR was conducted using TaqMan gene expression assays and the TaqMan Fast Advanced Master Mix (Life Technologies, Darmstadt, Germany) with 200 ng cDNA template on a StepOnePlus instrument (Life Technologies, Darmstadt, Germany). Assays were run in total volumes of 20 μL on 96-well plates (Sarstedt, Nümbrecht, Germany) and the following TaqMan gene expression assays were used: KRT20 (CK20), Hs00966063_m1; DEFA5, Hs00360716_m1; and TBP (TATA box binding protein), Hs00427621_m1 as a reference/housekeeping probe. All samples were run in triplicate. The mean threshold cycles of triplicate reactions were computed using the StepOne software v. 2.1 (Life Technologies, Darmstadt, Germany) after adjustment to the same threshold of all runs for each TaqMan assay on different plates. Gene expression was calculated as arbitrary expression units by a simplified ΔCt method normalizing the CK20 expression against the reference gene TBP [38].

### 3.4. Statistical Analysis

All of the clinical parameters were analyzed as a whole and then separately for tumor location and histopathological staging. To determine overall survival (OS), Kaplan–Meier survival analyses were implemented. As a means of assessing statistical significance with regard to univariate survival analysis, the log-rank test was used. In order to establish a correlation between CK20 detection rate and clinical parameters, we utilized the χ^2^ test after crosstab examination. We included those variables that significantly correlated with the detection of a biomarker in univariate analysis in multivariate models. For that, we used Cox proportional hazard models. Determination of cut-off expression values for CK20 and DEFA5 from qPCR data were calculated from receiver-operating-characteristics (ROC-)curves and the computed Youden index values [39]. All reported *p*-values are two-sided and were regarded statistically significant at *p* ≤ 0.05. For statistical calculation, testing and ROC curve analysis were performed with IBM SPSS Statistics 23.0 (IBM, München, Germany).

## 4. Discussion

Since there are still only a few studies [19,40] analyzing the detection of CTC in the blood and DTC in the bone marrow of EC patients, the objective of this study was to analyze the prognostic value of CK20 and DEFA5 mRNA expression levels in blood and bone marrow in a cohort of 216 EC patients. Although the expression of both genes has been described predominantly in the gastrointestinal tract according to the Human Protein Atlas (https://www.proteinatlas.org/ (accessed on 24 February 2022)) [41], and RNA-sequencing data in the Cancer Genome Atlas (TCGA) database indicate that the expression of either gene is not altered in EC compared to the normal tissue [42], CK20 was chosen as a biomarker for EC cells, because it has already been proven as a good indicator for CTC and DTC in CRC and EC patients with a high prognostic and predictive value [19,21,32]. Furthermore, we intended to validate our previous study, in which we analyzed blood and bone marrow samples from a smaller cohort of 77 EC patients [19].

To further validate this marker for detection of CTC and DTC in EC patients and with this as an indicator of malignancy, CK20 expression was also assessed in the blood and bone marrow of a control cohort (comprising healthy blood donors and bone marrow donors with non-malignant diseases) showing a positive detection of CK20 in the blood (63%) and bone marrow (42%). Our findings are in line with studies that have also demonstrated CK20 positivity in the blood of healthy donors [32]. Importantly, it has to be taken into account that all RT-PCR analyses were performed in triplicate with the samples from the EC patients being more often triple or at least double positive for CK20 (but also for DEFA5) than the control cohort. In this study, expression levels were used irrespective of the number of positive PCR samples. Considering the fact that blood samples of healthy donors were more often single positive than double or triple positive for CK20 and the cohort of EC patients comprised more blood and bone marrow samples with clearly higher expression levels of CK20 (but also DEFA5) compared to the control samples, it can be speculated that the number of CK20-positive MNC is much lower in healthy donors than in EC patients. However, in order to consider this background expression and to clearly identify malignancy-associated positive results, a threshold value was calculated. Applying the calculated threshold value, the detection rate of CK20 was 22.5% in the blood of EC patients, which corresponds to the lower range of the literature, where detection rates of 19.7–57.4% are described [40,43]. The detection rate of CK20 expression in the bone marrow of EC patients at 33.3% was also in line with the literature, e.g., Pantel et al., could detect DTC by immunocytochemical analyses in the bone marrow in 20–40% of affected patients [44].

As another promising biomarker, DEFA5, was chosen for this study. Although DEFA5 has already been shown to have a cancer-protective function in gastric and colon cancer, [32,45], its role in tumorigenesis is still incompletely understood, especially regarding EC. However, elevated DEFA5 expression or serum levels were found in association with CRC [28,29]. Moreover, Born et al., could demonstrate elevated DEFA5 mRNA expression in blood-derived MNC of CRC patients compared to healthy donors [32]. To the best of our knowledge, the present study is the first to investigate the presence of DEFA5 in blood or bone marrow and its prognostic value in EC patients. 

Besides their antimicrobial activity, defensins have been reported to play a role in tumorigenesis and host defense against tumors [46,47,48,49]. As part of the α-defensin subfamily, DEFA5 is produced and discharged by Paneth cells, which are secretory epithelial cells in the small intestine [45]. In addition, human neutrophil defensins are constitutively synthesized in promyelocytes and early myelocytes in the bone marrow [50,51]. After their maturation, defensins are released into the blood. Therefore, detection of DEFA5 mRNA in blood and bone marrow also in the control group is comprehensible, implicating that a calculation of a threshold value is required to use DEFA5 as a prognostic biomarker. 

After calculation and applying the respective cut-off value, the detection rate of DEFA5 was refined from 70.8% to 45.9% in the blood and from 100% to 66.7% in the bone marrow of EC patients. In the control group, 57.8% of the blood and 91.6% of the bone marrow samples were positive for DEFA5, which declined to 13% and 45%, respectively, after cut-off calculation. The higher detection rates of DEFA5 in the blood of EC patients compared to healthy controls correspond to the findings for CRC patients published by Born et al., and the findings by Lisitsyn et al., albeit the latter study comprises only a little number of patients and control donors [28,32]. 

On the other hand, the work of Wu et al., showed that a drastic downregulation of DEFA5 can be detected in patients with gastric carcinoma. These findings along with experimental data demonstrating a tumor cell growth inhibitory effect of DEFA5 point to a role of DEFA5 as a tumor suppressor [34,35]. Considering its tumor-suppressive function and its known immunomodulating role of DEFA5, the detection of DEFA5 even in healthy donors is reasonable and has also been reported earlier [32]. Moreover, it may also explain why DEFA5 positivity in blood (EC patients without neoadjuvant therapy) and bone marrow (EC patients with neoadjuvant therapy) is associated with a better outcome of EC patients. 

However, overall, no correlation between detection of CK20 and DEFA5, respectively, in the blood and bone marrow of EC patients and UICC stage, TNM category, gender, histological tumor type, adjuvant or neoadjuvant chemotherapy was observed.

In contrast to some studies [43,52,53,54], we could also not show a significant correlation between the detection of either biomarker in the blood or bone marrow and OS of our total and non-subdivided patient collective of 216 EC patients. This might be explained by the fact that this cohort comprises patients with and without neoadjuvant therapy, the latter being able to eliminate and affect, respectively, EC cells in liquid biopsies or at least certain subtypes thereof.

However, regarding CK20 mRNA expression, we could demonstrate that CK20 positivity in bone marrow is associated with shorter OS in EC patients without neoadjuvant therapy. Since we could not observe any correlation between CK20 mRNA expression and OS in EC patients with neoadjuvant treatment, it can be speculated that neoadjuvant therapy either eliminated CK20 positive CTC and DTCs or promoted an epithelial–mesenchymal transition (EMT) by which carcinoma cells can lose epithelial markers such as cytokeratin [55]. In line with these findings, studies could demonstrate that CTC lacks expression of epithelial markers while expressing mesenchymal markers. Accordingly, the difficulty of capturing this subgroup of cells by using epithelial markers for CTC/DTC enrichment and detection, respectively, might explain why detection of those markers (in our study CK20) does not necessarily correlate with survival and prognosis of cancer patients, as it has been shown, e.g., for ovarian cancer [56].

Another reason for the lack of correlation between detection of CK20 and DEFA5 and OS may be the heterogeneity of our overall cohort which comprises patients with and without neoadjuvant therapy. Therefore, the total patient collective was divided into patients with neoadjuvant therapy and without neoadjuvant treatment for a stratified analysis.

Analysis of a sub-collective of 101 patients without neoadjuvant therapy revealed a clearly significant correlation between bone marrow CK20 positivity and overall survival, which was different compared to the analysis of the entire collective. Thus, after five years, only 18% of the patients with CK20-positive bone marrow were still alive in the cohort of patients without neoadjuvant therapy in contrast to 54% of the patients who were CK20 negative. Furthermore, the multivariate Cox regression analysis revealed that CK20 positivity in the bone marrow is an independent prognostic factor for OS of EC patients who have not received neoadjuvant therapy. This finding is in line with other studies demonstrating a negative correlation of DTC in bone marrow and survival in patients with different tumor entities, such as EC, breast cancer, lung cancer, colorectal, and pancreatic carcinoma [57,58,59].

Further examination of our sub-collective interestingly revealed that there is also a significantly positive correlation between DEFA5 detection in the blood and OS of patients who did not receive a neoadjuvant therapy. Patients with DEFA5-positive blood samples survived significantly longer (48% 5-year survival rate) than patients with DEFA5 negativity in the blood (30% 5-year survival rate). In line with these findings, Wu et al., described overexpression of DEFA5 in gastric carcinoma leading to diminished cell proliferation suggesting a role of DEFA5 as a tumor suppressor [35].

Overall, these data indicate that CK20 positivity in bone marrow or DEFA5 positivity in blood is of prognostic value for EC patients without neoadjuvant therapy. Therefore, it may be feasible to monitor the DTC or CTC values in EC patients during and after therapy, as it has already been done in breast cancer patients, as a means of gauging responses to therapy. Moreover, it might also facilitate the detection of recurrences at an earlier stage than conventional imaging techniques [60,61].

Furthermore, DTC/CTC detection might also be helpful for personalized treatment or multimodal concepts. According to the current guidelines of the German Cancer Society, patients with a low tumor stage and no lymph node metastases commonly do not receive multimodal therapy [62]. However, if there is evidence of DTC in these patients, a multimodal therapy concept might be indicated and adjusted after surgery improving the survival of these patients [63,64].

By examining the sub-collective of 115 patients who received neoadjuvant chemotherapy, of which *n* = 82 BM samples were analyzed, DEFA5 positivity in the bone marrow was significantly associated with a longer OS (5-year OS of 47% in patients with DEFA5 positivity in the BM versus 5-year OS of 16% in patients with DEFA5-negative BM). The results on DEFA5 expression and its association with better OS give reason to assume that it may function as a tumor suppressor, which is in line with previous findings on the growth inhibitory effect of DEFA5 in gastric cancer [35]. Therefore, further research is needed to better understand the role of DEFA5 in the development and progression of EC and to implement these findings into clinical practice. 

As previously observed, neoadjuvant chemotherapy partially eradicates DTC in the bone marrow [65], which may explain the above-mentioned correlation. Accordingly, the prognosis for patients who do not express DEFA5 in their bone marrow following chemotherapy is poor. These findings hold a high value for future therapy concepts because those are precisely the patients requiring detection at an early stage. 

## 5. Conclusions

To the best of our knowledge, our work is the first to investigate the prognostic value of CK20 and DEFA5 expression in liquid biopsies of a large cohort of 216 EC patients.

Thus, our study revealed the prognostic value of DEFA5 expression in the blood and bone marrow of EC patients. In patients without neoadjuvant therapy, the absence of CK20 in bone marrow and DEFA5 positivity in blood was associated with an improved OS. In addition, the patients with a positive DEFA5 detection in the bone marrow showed a longer OS in the group of EC patients with neoadjuvant therapy. Thus, these data demonstrate that biomarker detection in bone marrow and blood might have a different impact on OS in EC patients indicating a divergent biological significance of CTC or DTC which is still not fully understood. However, our data point to the potential of DEFA5 as a suitable prognostic biomarker in liquid biopsies of EC patients. Of note, bone marrow blood collection is an invasive procedure not necessarily required for EC treatment and is associated with a higher risk of complications compared to peripheral blood collection so its implementation in clinical routine must be considered carefully. However, given the fact there is an acute need for reliable biomarkers in MRD monitoring of EC patients to better assess the effect of neoadjuvant and adjuvant therapy, future studies involving the longitudinal collection of liquid biopsies should be performed to elaborate whether this marker is suitable for disease monitoring and therapy decision making.

## Figures and Tables

**Figure 1 cancers-14-01279-f001:**
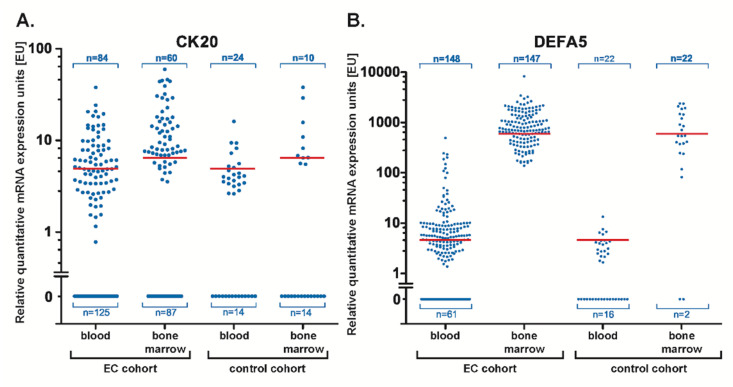
Relative quantitative expression levels of CK20 and DEFA5 mRNA in blood and bone marrow of patients with esophageal carcinoma (EC) or non-malignant diseases and healthy controls (control cohort). Blood samples of 209 EC patients and 38 benign control patients, bone marrow samples of 147 EC patients, and 24 control patients with non-malignant diseases were semi-quantitatively analyzed for the expression of (**A**) CK20 and (**B**) DEFA5 mRNA. The determined cut-off values were calculated to be 4.64 relative mRNA expression units (EU) for CK20 in blood and 6.06 EU for CK20 in bone marrow. For DEFA5, the cut-off was 4.26 EU in blood and 599.3 EU in bone marrow. These values are shown as red lines in the graph.

**Figure 2 cancers-14-01279-f002:**
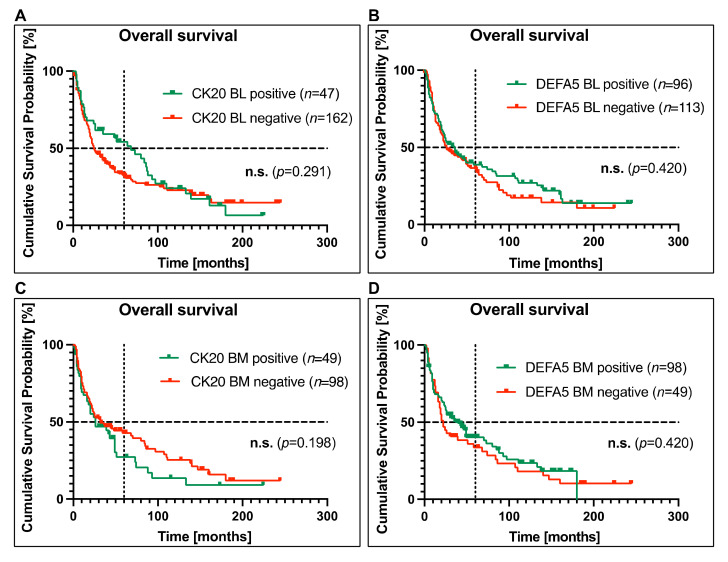
Correlation between CK20 and DEFA5 positivity in blood and bone marrow samples, respectively, and survival for patients with esophageal carcinoma (EC). Detection of (**A**,**C**) CK20 and (**B**,**D**) DEFA5 in (**A**,**B**) blood and (**C**,**D**) bone marrow samples of EC patients was assessed by RT-PCR and correlated with overall survival by Kaplan–Meier analysis. On the vertical dotted line, the 5-year (60 months) overall survival is shown. The horizontal dotted line at the cumulative survival probability of 0.5 indicates the median survival. (BL: blood, BM: bone marrow, n.s.: not significant).

**Figure 3 cancers-14-01279-f003:**
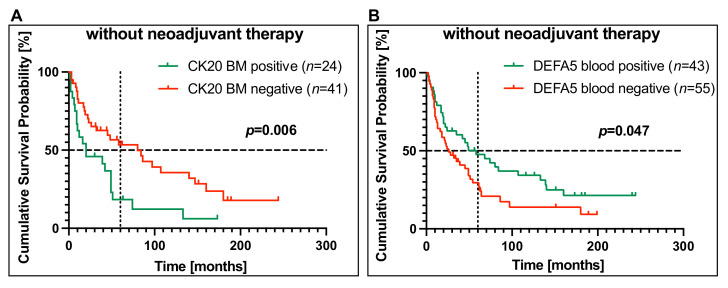
Correlation of CK20 in bone marrow and DEFA5 in blood with cumulative overall survival in esophageal carcinoma (EC) patients without neoadjuvant chemotherapy. (**A**) CK20 expression in bone marrow samples (BM) (*n* = 65) and (**B**) DEFA5 expression in blood samples (*n* = 98) of EC patients being therapy-naive before surgery was assessed by RT-PCR and correlated with overall survival by Kaplan–Meier analysis. On the vertical dotted line, the 5-year (60 months) overall survival is shown. The horizontal dotted line at the cumulative survival probability of 0.5 indicates the median survival. (BL: blood, BM: bone marrow).

**Figure 4 cancers-14-01279-f004:**
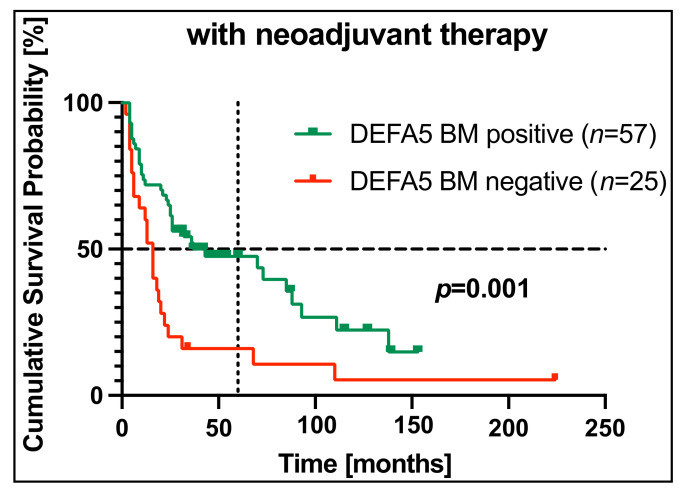
Correlation of DEFA5 in bone marrow and survival in esophageal carcinoma (EC) patients with neoadjuvant chemotherapy. DEFA5 expression in bone marrow (BM) (*n* = 82) samples of EC patients with neoadjuvant therapy was assessed by RT-PCR and correlated with overall survival. On the vertical dotted line, the 5-year (60 months) overall survival is shown. The horizontal dotted line at the cumulative survival probability of 0.5 indicates the median survival. (BL: blood, BM: bone marrow).

**Table 1 cancers-14-01279-t001:** Patients’ clinical and pathological characteristics and the corresponding *p*-values of log-rank tests of univariate Kaplan–Meier analyses of the correlation of these variables with the 5-year overall survival rate (OS). *p*-values indicating a correlation are printed in bold.

	Category	*n* (%)	5y-OS (%)	*p*-Value
All		216 (100)	38	
Gender	Male	182 (84.3)	37	0.95
	Female	34 (15.7)	43	
Age [years]	<70	160 (74.1)	42	**0.003**
	>70	56 (25.9)	26	
Histotype	Adenocarcinoma	162 (75)	35	0.141
	Squamous cell carcinoma	54 (25)	42	
pT category (*n* = 200)	T1	55 (27.5)	65	**<0.001**
	T2	50 (25)	19	
	T3	93 (46.5)	28	
	T4	2 (1)	0	
pN category	N0	105 (48.6)	58	**<0.001**
	N1	73 (33.8)	23	
	N2	29 (13.4)	13	
	N3	9 (4.2)	0	
pM category	M0	201 (93.1)	40	**0.001**
	M1	15 (6.9)	7	
UICC stage (*n* = 200)	I	48 (24)	65	**<0.001**
	II	52 (26)	37	
	III	77 (38.5)	29	
	IV	23 (11.5)	4	
Neoadjuvant therapy	Yes	115 (53.2)	39	0.323
	No	101 (46.8)	38	
Adjuvant therapy	Yes	57 (26.4)	41	0.107
	No	138 (63.9)	37	
	Unknown	21 (9.7)		

UICC: Union Internationale Contre le Cancer.

**Table 2 cancers-14-01279-t002:** CK20 and DEFA5 mRNA detection rates above the individual cut-off values in blood or bone marrow of patients with esophageal carcinoma (EC). Blood samples of 209 EC patients and bone marrow samples of 147 EC patients were analyzed for the presence of CK20 and DEFA5 mRNA.

Variables	Blood Samples (*n* = 209) (%)		Bone Marrow Samples (*n* = 147) (%)	
	Positive (%)	Negative (%)	Positive (%)	Negative (%)
CK20	47 (22.5)	162 (77.5)	49 (33.3)	98 (66.7)
DEFA5	96 (45.9)	113 (54.1)	98 (66.7)	49 (33.3)

**Table 3 cancers-14-01279-t003:** Number of esophageal carcinoma patients with detectable expression of CK20 (CK20+) and DEFA5 (DEFA5+) in blood and bone marrow and their correlation with clinical parameters.

Variable		CK20 + BL (%)	*p*	DEFA5 + BL (%)	*p*	CK20 + BM (%)	*p*	DEFA5 + BM (%)	*p*
Gender	Male	38 (21.5)	0.40	83 (46.9)	0.51	44 (34.6)	0.39	85 (66.9)	0.86
	Female	9 (28.1)		13 (40.6)		5 (25)		13 (65)	
Age [years]	<70	32 (20.6)	0.28	72 (46.5)	0.79	34 (30.9)	0.28	74 (67.3)	0.79
	>70	15 (27.8)		24 (44.4)		15 (40.5)		24 (64.9)	
Histotype	AC	35 (22.6)	0.78	70 (45.2)	0.71	39 (34.5)	0.50	78 (69)	0.25
	SCC	12 (22.2)		26 (48.1)		10 (29.4)		20 (58.8)	
pT category	T1	12 (22.2)	0.81	29 (53.7)	0.30	12 (33.3)	0.54	23 (63.9)	0.24
	T2	12 (25.5)		21 (44.7)		15 (39.5)		30 (78.9)	
	T3	19 (20.9)		38 (41.8)		18 (29)		37 (59.7)	
	T4	0		0		0		1 (50)	
pN category	N0	25 (24.8)	0.73	52 (51.5)	0.24	22 (31.4)	0.79	49 (70)	0.55
	N1	13 (18.1)		32 (44.4)		19 (38)		32 (64)	
	N2	7 (25.9)		10 (37)		7 (31.8)		15 (68.2)	
	N3	2 (22.2)		2 (22.2)		1 (20)		2 (40)	
pM category	M0	45 (23.2)	0.38	88 (45.4)	0.55	47 (34.6)	0.27	91 (66.9)	0.82
	M1	2 (13.3)		8 (53.3)		2 (18.2)		7 (63.6)	
UICC stage	I	11 (23.4)	0.91	27 (57.4)	0.30	11 (35.5)	0.91	21 (67.7)	0.79
	II	10 (20)		21 (42)		12 (32.4)		26 (70.3)	
	III	18 (24)		31 (41.3)		18 (33.3)		35 (64.8)	
	IV	4 (18.2)		9 (40.9)		4 (25)		9 (56.3)	
Neoadjuvant therapy	Yes	21 (18.9)	0.19	53 (47.7)	0.57	25 (30.5)	0.41	57 (69.5)	0.41
	No	26 (26.5)		43 (43.9)		24 (36.9)		41 (63.1)	
Adjuvant therapy	Yes	8 (14.8)	0.08	21 (38.9)	0.31	13 (31.7)	0.73	32 (78)	0.07
	No	36 (26.9)		63 (47)		32 (34.8)		57 (62)	

BL: blood, BM: bone marrow, AD: adenocarcinoma, SCC: squamous cell carcinoma, UICC: Union Internationale Contre le Cancer.

**Table 4 cancers-14-01279-t004:** Multivariate Cox regression analysis of independent factors and correlation with overall survival of esophageal carcinoma patients (all *p*-values in bold are regarded as statistically significant, BM: bone marrow, BL: blood, HR: hazard ratio, CI: confidence interval).

Variables	Overall Survival	
	Univariate	Multivariate
	*p*-Value	HR (95% CI) *p*-Value
**Without neoadjuvant therapy**		
CK20 detection in BM	0.006	**2.28 (1.24–4.22) 0.008**
DEFA5 detection in BL	0.047	0.74 (0.41–1.34) 0.318
UICC	<0.001	**1.60 (1.19–2.17) 0.002**
**With neoadjuvant therapy**		
DEFA5 detection in BM	0.001	**0.46 (0.26–0.80) 0.006**
UICC	<0.001	**1.53 (1.10–2.13) 0.012**
DEFA5 detection in BM	0.001	**0.46 (0.26–0.80) 0.006**

## Data Availability

Data is contained within this article.

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
