# Peer review of "Detection of Circulating and Disseminated Tumor Cells and Their Prognostic Value under the Influence of Neoadjuvant Therapy in Esophageal Cancer Patients"

_cancers, 2022, doi:10.3390/cancers14051279_

Round 1

Reviewer 1 Report

The following points could be included in your discussion:

  1. RNA-seq data in the TCGA database show that none of these genes has altered expression in esophageal adenocarcinoma compared to the normal tissue.
  2. Both KRT20 and DEFA5 mRNA have highest expression in normal gastrointestinal track according to The Human Protein Atlas.

Author Response

Dear reviewer, 

thank you very much again for the rapid evaluation of our revised manuscript entitled “Detection of circulating and disseminated tumor cells and their prognostic value under the influence of neoadjuvant therapy in esophageal cancer patients” and the helpful suggestion for further improving our manuscript.

According to the your comment, we have further modified our revised version of the manuscript that you will find enclosed with this letter. All changes have been highlighted by underlining.

Below, we will explain in detail how the suggestion has been dealt with:

Request: The following points could be included in your discussion:

  1. RNA-seq data in the TCGA database show that none of these genes has altered expression in esophageal adenocarcinoma compared to the normal tissue.
  2. Both KRT20 and DEFA5 mRNA have highest expression in normal gastrointestinal track according to The Human Protein Atlas

Answer: According tot he suggestion oft he reviewer we have included the following text passage in the discussion on page 11/12: „Although expression of both genes has been described predominantly in the gastrointestinal tract according to the Human Protein Atlas (https://www.proteinatlas.org/ (accessed on 24th February 2022)) [41] and RNA-sequencing data in the Cancer Genome Atlas (TCGA) database indicate that expression of either gene is not altered in EC compared to the normal tissue [42], CK20 was chosen as biomarker for EC cells, because it has already been proven as a good indicator for CTC and DTC in CRC- and EC patients with a high prognostic and predictive value [19,21,32]. Furthermore, we intended to validate our previous study, in which we analyzed blood and bone marrow samples from a smaller cohort of 77 EC patients [19].“

We very much hope that these additions sufficiently improved our paper making it now suitable for publication in Cancers.

Thank you very much for your consideration and efforts!

Sincerely yours,

S. Sebens, PhD

This manuscript is a resubmission of an earlier submission. The following is a list of the peer review reports and author responses from that submission.

Round 1

Reviewer 1 Report

  1. Review report to the Authors:

In this study, Richter et al. analyze mRNA levels of two proposed markers in EC patients in blood and bone marrow. The authors conduct extensive qPCR measurements to identify specific mRNA levels in CTCs from patient sera and correlate these with clinical parameters.

The main findings of the manuscript are: (1) enhanced BM CK20 / reduced blood DEFA5 levels correlated with reduced survival in untreated patients, and (2) enhanced levels of BM DEFA5 correlated with improved survival in patients undergoing neoadjuvant therapy. Rather surprisingly, the authors find DEFA5 expression associated with enhanced survival, though it was described as an EMT and, hence, tumor spread promoting factor. The authors point out the prognostic relevance of this factor.

Analyses and experiments conducted in the manuscript are extensive, they follow robust methods and are well controlled. The argumentation is concise, the manuscript is clearly structured, and the results are described highly detailed. This is a sound study. However, the authors should be cautious in making strong interpretation when comparing expression of genes that are at low end of expression values due to unclear cellular origin. I recommend publication after some minor changes.

Minor comments:

1) The authors use MNCs as source of their RNA screens, assuming that these also contain tumor derived CTCs/DTCs. It would be important to clearly characterize the identity of these cells, possibly with certain other markers. How can the authors be sure that the two markers are indeed tumor derived and tumor relevant?

2) Technical: RNA quantification with nano drop is not very accurate and could be improved with other methods e.g. Agilent Bioanalyzer. With technical refinements, therefore, much more accurate and meaningful results could be obtained.

3) Discussion: I would be cautious with arguing on the one hand weak CK20 detection levels caused by EMT indicating tumor aggressiveness, and on the other hand correlate increased CK20 levels with reduced survival. Please interpret and discuss consistently.

4) The authors found increased BM CK20 and reduced blood DEFA5 mRNA levels in untreated EC patients correlating with reduced OS. It is a little confusing when the results are described to opposite survival outcomes, especially regarding the DEFA5 results. It would be easier to follow if the correlation was described consistently, e.g. reduced DEFA5 always in relation to decrased survival.

Author Response

Dear reviewer 1,

thank you very much for the rapid evaluation of our manuscript entitled “Detection of circulating and disseminated tumor cells and their prognostic value under the influence of neoadjuvant therapy in esophageal cancer patients” and the helpful suggestions for improving our manuscript.

According to the reviewer`s comments, we have now prepared a revised version of the manuscript that you will find enclosed with this letter. All changes have been highlighted by underlining.

Below, we will explain point-by-point how the arguments and criticisms of the reviewer have been dealt with:

Request 1: The authors use MNCs as source of their RNA screens, assuming that these also contain tumor derived CTCs/DTCs. It would be important to clearly characterize the identity of these cells, possibly with certain other markers. How can the authors be sure that the two markers are indeed tumor derived and tumor relevant?

Answer 1: We agree that this is an improtant issue. We have incorporated the following sentence along with two references in the introduction on page 2: „MNC enrichment by Ficoll density centrifugation represents a long-established technique for depleting possibly CK20-positive granulocytes from whole blood samples [23], also enriching CTC and DTC as demonstrated in numerous studies with clinical samples as well as in preanalytical validation experiments with spiked tumor cells in whole blood [24]. In the present study, gene expression was analyzed by using reverse-transcription realtime-polymerase chain reaction (RT-PCR)-assays.“

Request 2:  Technical: RNA quantification with nano drop is not very accurate and could be improved with other methods e.g. Agilent Bioanalyzer. With technical refinements, therefore, much more accurate and meaningful results could be obtained.

Answer 2: We thank the reviewer for pointing out the limited accuracy of the Nanodrop measurement. As our goal was not an absolute quantification of mRNA expression, rather than a relative detection of marker expression in relation to a housekeeping marker, we think that this limited accuracy is acceptable for this study. Importantly, all Nanodrop measurements were in a similar range for all total RNA samples and all realtime RT-PCR Ct-values for the housekeeper (TBP) were highly similar.

Request 3:  Discussion: I would be cautious with arguing on the one hand weak CK20 detection levels caused by EMT indicating tumor aggressiveness, and on the other hand correlate increased CK20 levels with reduced survival. Please interpret and discuss consistently.

Answer 3: We agree with the reviewer that this issue requires a better interpretation and discussion. Accordingly, we have thoroughly rewritten this part of the discussion. We have modified as follows: „In contrast to some studies [41,50-52], we could also not show a significant correla-tion between the detection of either biomarker in the blood or bone marrow and OS of our total and not subdivided patient collective of 216 EC patients. This might be explained by the fact that this cohort comprises patients with and without neoadjuvant therapy, the latter being able to eliminate and affect, respectively, EC cells in liquid biopsies or at least certain subtypes thereof. However, regarding CK20 mRNA expression we could demonstrate that CK20 pos-itivity in bone marrow is associated with shorter OS in EC patients without neoadjuvant therapy. Since we could not observe any correlation between CK20 mRNA expression and OS in EC patients with neoadjuvant treatment it can be speculated that neoadjuvant therapy either eliminated CK20 positive CTC and DTCs or promoted an epitheli-al-mesenchymal-transition (EMT) by which carcinoma cells can lose epithelial markers such as cytokeratins [53]. In line with these findings, studies could demonstrate that CTC lack expression of epithelial markers while expressing mesenchymal markers. Accord-ingly, the difficulty of capturing this subgroup of cells by using epithelial markers for CTC/DTC enrichment and detection, respectively, might explain why detection of those markers (in our study CK20) does not necessarily correlate with survival and prognosis of cancer patients, as it has been shown e.g. for ovarian cancer [54].“

Request 4:  The authors found increased BM CK20 and reduced blood DEFA5 mRNA levels in untreated EC patients correlating with reduced OS. It is a little confusing when the results are described to opposite survival outcomes, especially regarding the DEFA5 results. It would be easier to follow if the correlation was described consistently, e.g. reduced DEFA5 always in relation to decrased survival.

Answer 4: As suggested and in seek of clarity, we have adapted the descriptions of CK20 and DEFA5 expression in liquid biospies and its correlation with OS accordingly throughout the manuscript.

We very much hope that these modifications and additions sufficiently improved our paper making it now suitable for publication in Cancers.

Thank you very much for your consideration and efforts!

Sincerely yours,

S. Sebens, PhD

Reviewer 2 Report

This study determines the prognostic value of cytokeratin 20 (CK20) and alpha-defensin 5 (DEFA5) for detection of circulating tumor cells (CTC) and disseminated tumor cells (DTC) from esophageal cancer (EC). The authors collected blood and bone marrow (BM) samples from 216 EC patients with or with undergone neoadjuvant therapy. They also obtained blood and BM samples from healthy individuals or subjects with benign conditions as controls. The authors used real-time RT-PCR to measure the expression levels of CK20 and DEFA5 and performed a number of analyses to evaluate their association with the survival of the EC patients, with or without taking neoadjuvant therapy into consideration. The authors show that while CK20 and DEFA5 expression in the blood or BM have no prognostic value for EC when all patients are pooled together, the expression of CK20 in BM is negatively associated while that of DEFA5 in the blood is positively associated with the survival of EC patients without neoadjuvant therapy. The authors conclude that CK20 positivity in BM and DEFA5 negativity in the blood have prognostic value for EC patients without neoadjuvant therapy. This is a reasonably conduced, albeit somewhat limited, study. Data collection, data analyses, and conclusions are mostly appropriate. The manuscript is generally well-written. However, some weaknesses discussed below need to be addressed.

  1. This is a rather limited study looking at two potential markers for EC. It would be helpful to include additional and more established markers for EC in the study to better evaluate these two genes.
  2. It would be helpful to include some clinical data from cancer databases (e.g. TCGA) in the analysis and/or discussion regarding the prognostic value CK20 and DEFA5.
  3. It is a little confusing as what control group is. Page 4, line 142 describes the control being “from patients with benign diseases” whereas page 10, line 328 states the control being “healthy volunteers” and line 331 says “24 patients suffering from non-malignant diseases”. If the blood samples were from healthy individuals and BM samples were from patients with benign conditions, would some of the conclusions be affected if the controls were all from healthy subjects?
  4. The p value for DEFA5 in the blood of patients without adjuvant therapy (Fig. 3B) is at the border line for significance while that of DEFA5 in BM of patients with neoadjuvant therapy is 0.001 (Fig. 4). Why DEFA5 in BM is not a better prognostic marker for patients with neoadjuvant therapy? The explanation feels inadequate.
  5. Given that DEFA5 is highly expressed in the esophageal epithelium (and possibly EC?), the reason underlying the positive rather than negative association of DEFA5 with overall survival of the EC patients is puzzling. More discussion is needed.
  6. There are no differences in the levels CK20 and DEFA5 between samples from EC and control groups (Fig. 1). This diminishes the overall value of these two genes as prognostic markers.

Author Response

Dear reviewer 2,

thank you very much for the rapid evaluation of our manuscript entitled “Detection of circulating and disseminated tumor cells and their prognostic value under the influence of neoadjuvant therapy in esophageal cancer patients” and the helpful suggestions for improving our manuscript.

According to the reviewer`s comments, we have now prepared a revised version of the manuscript that you will find enclosed with this letter. All changes have been highlighted by underlining.

Below, we will explain point-by-point how the arguments and criticisms of the reviewer have been dealt with:

The manuscript is generally well-written. However, some weaknesses discussed below need to be addressed.

Request 1: This is a rather limited study looking at two potential markers for EC. It would be helpful to include additional and more established markers for EC in the study to better evaluate these two genes.

Answer 1: First of all, we would like to point out that based on our former study demonstrating the prognostic value of CK20 in liquid biopsies in a small corhort of EC patients, the aim of this study was to validate this finding and to assess the suitability of this marker in a larger cohort comprising 216 EC patients. Importantly, all other cited studies investigating CTC/DTC in liquid biopsies of EC patients were much smaller. Secondly, DEFA5 was additionally investigated because of its putative role in gastrointestinal tumorigenesis and as we could recently demonstrate that DEFA5 expression level are elevated in blood derived MNC fractions of colorectal cancer patients providing the rationale to investigate this marker also in liquid biopsies of EC patients. This piece of information along with the reference have been included in the introduction on page 3 to underscore the rationale for investigating particularly these two markers.

Request 2: It would be helpful to include some clinical data from cancer databases (e.g. TCGA) in the analysis and/or discussion regarding the prognostic value CK20 and DEFA5.

Answer 2: In general we agree with the reviewer that clinical data from a cancer database might be helpful for assessing the prognostic value of the two biomarkers. However, since we did not analyze mutations or other genetic alterations, we do not consider TCGA data to be of further use in our present study. Moreover, to the best of our knowledge, our work is the first investigating the prognostic value of CK20 and DEFA5 expression in liquid biopsies of a large cohort of 216 EC patients. Accordingly, data on the expression of either marker in liquid biospies of EC patients are not existing or rare and are cited in our manuscript.

Request 3: It is a little confusing as what control group is. Page 4, line 142 describes the control being “from patients with benign diseases” whereas page 10, line 328 states the control being “healthy volunteers” and line 331 says “24 patients suffering from non-malignant diseases”. If the blood samples were from healthy individuals and BM samples were from patients with benign conditions, would some of the conclusions be affected if the controls were all from healthy subjects?

Answer 3: We agree with the reviewer that this might be confusing. Hence, for uniformity and clarity, the manuscript text was changed on page 4 into: „38 blood samples from healthy donors and 24 bone marrow samples from patients with non-malignant diseases“. Accordingly, the manuscript text in the „Materials and Methods“ section on page 10 was also changed into: „Furthermore, bone marrow samples were collected from 24 patients suffering from non-malignant diseases including one patient with liver cysts, three patients from whom parts of multiorgan donations were obtained, five patients suffering from pulmonary hematoma, three patients with hepatic hemangioma, three patients suffering from focal nodular hyperplasia of the liver, five patients with sigmoid diverticulitis and four patients with chronic pancreatitis.“

Request 4: The p value for DEFA5 in the blood of patients without adjuvant therapy (Fig. 3B) is at the border line for significance while that of DEFA5 in BM of patients with neoadjuvant therapy is 0.001 (Fig. 4). Why DEFA5 in BM is not a better prognostic marker for patients with neoadjuvant therapy? The explanation feels inadequate.

Answer 4: According to the suggestion of the reviewer we have discussed in more detail the findings of DEFA5 expression in bone marrow in the discussion on page 14 (and please see also our response to request 5): „However, our data point to the potential of DEFA5 as a suitable prognostic biomarker in liquid biopsies of EC patients. Of note, bone marrow blood collection is an invasive procedure not necessarily required for EC treatment and associated with a higher risk of complications compared to peripheral blood collection so that its implementation in clinical routine has be considered carefully. However, given the fact there is an acute need for reliable biomarkers in MRD monitoring of EC patients to better assess the effect of neoadjuvant and adjuvant therapy, future studies involving longitudinal collection of liquid biopsies should be performed to elaborate whether this marker is suitable for disease monitoring and therapy decision making.“

Request 5: Given that DEFA5 is highly expressed in the esophageal epithelium (and possibly EC?), the reason underlying the positive rather than negative association of DEFA5 with overall survival of the EC patients is puzzling. More discussion is needed.

Answer 5: We agree with the reviewer that this issue deserves an explanation. Therefore, we have incorporated the following text in the discussion on page 14: „By examining the sub-collective of 115 patients who received neoadjuvant chemotherapy, of which n=82 BM samples were analyzed, DEFA5 positivity in the bone marrow was significantly associated with a longer OS (5-year OS of 47% in patients with DEFA5 positivity in the BM versus 5-year OS of 16% in patients with DEFA5-negative BM). The results on DEFA5 expression and its association with better OS give reason to assume that it may function as a tumor suppressor being in line with previous findings on the growth inhibitory effect of DEFA5 in gastric cancer [35]. Therefore, further research is needed to better understand the role of DEFA5 in development and progression of EC and to implement these findings into clinical practice.“

Request 6: There are no differences in the levels CK20 and DEFA5 between samples from EC and control groups (Fig. 1). This diminishes the overall value of these two genes as prognostic markers.

Answer 6: We have to appologize to having not make this more clear to the readership. Accordingly, we have thoroughly modified chapter 2.2 and 2.3 in the results section:

„Thus, both markers examined were also detected in the control cohort. However, it has to be taken into account that all RT-PCR analyses were performed in triplicates with the samples from the EC patients being more often triple or at least double positive for either marker than the control cohort.“

„In the control cohort, expression levels in blood ranged from 2.46 EU to 15.15 EU for CK20 and from 2.15 EU to 13.47 EU for DEFA5. Examination of bone marrow samples showed expression units for CK20 in the range of 5.26 EU to 35.69 EU and for DEFA5 of 81.42 EU to 2384.78 EU. Even though the cohort of EC patients comprised more blood and bone marrow samples with clearly higher expression levels of either marker compared to the control samples, a high variability of the expression values was determined, so that it was decided to determine cut-off values for sample positivity, …“.

And we have rewritten the discussion on this topic as follows:

„To further validate this marker for detection of CTC and DTC in EC patients and with this as an indicator of malignancy, CK20 expression was also assessed in blood and bone marrow of a control cohort (comprising healthy blood donors and bone marrow donors with non-malignant diseases) showing a positive detection of CK20 in blood (63%) and bone marrow (42%). Our findings are in line with studies which have also demonstrated CK20 positivity in blood of healthy donors [32]. Importantly, it has to be taken into account that all RT-PCR analyses were performed in triplicates with the samples from the EC patients being more often triple or at least double positive for CK20 (but also for DEFA5) than the control cohort. In this study expression levels were used irrespective of the number of positive PCR samples. Considering the fact that blood samples of healthy donors were more often single positive that double or triple positive for CK20 and the cohort of EC patients comprised more blood and bone marrow samples with clearly higher expression levels of CK20 (but also DEFA5) compared to the control samples, it can be speculated that the number of CK20 positive MNC is much lower in healthy donors than in EC patients. However, in order to consider this background expression and to clearly identify malignancy-associated positive results, a threshold value was calculated.“

We very much hope that these modifications and additions sufficiently improved our paper making it now suitable for publication in Cancers.

Thank you very much for your consideration and efforts!

Sincerely yours,

S. Sebens, PhD